# *Blastocystis* colonization and associations with population parameters in Thai adults

**Vasana Jinatham[1], Amara Yowang[2], Christen Rune Stensvold[3], Eleni Michalopoulou[4], Thanakrit Vichasilp[5], Picha Suwannahitatorn[5], Siam Popluechai[1,6]\*, Anastasios D. Tsaousis[7], Eleni Gentekaki[1,4,6]\***

**1** School of Science, Mae Fah Luang University, Chiang Rai, Thailand, **2** Regional Medical Sciences Center 1/1 Chiang Rai, Department of Medical Sciences, Ministry of Public Health, Chiang Rai, Thailand, **3** Department of Bacteria, Parasites and Fungi; Statens Serum Institut, Copenhagen, Denmark, **4** Department of Veterinary Medicine, University of Nicosia School of Veterinary Medicine, Nicosia, Cyprus, **5** Phramongkutklao College of Medicine, Bangkok, Thailand, **6** Gut Microbiome Research Group, Mae Fah Luang University, Chiang Rai, Thailand, **7** Laboratory of Molecular and Evolutionary Parasitology, RAPID group, School of Biosciences, University of Kent, Canterbury, United Kingdom

\* siam@mfu.ac.th (SP); gentekaki.ele@mfu.ac.th (EG)

## Abstract

### Background

*Blastocystis* is a unicellular eukaryote commonly found in the intestinal tract of humans and other animals. The prevalence of *Blastocystis* has been investigated in both developed and developing countries, yet its occurrence and distribution in rural locations has been less studied. Herein, we aimed to examine the distribution of *Blastocystis* colonization in Thai adults representing background populations along a rural/peri-urban gradient, as well as associations between colonization and personal characteristics.

### Methodology

A total of 238 participants were recruited from rural and peri-urban areas situated in three provinces. The presence of *Blastocystis* in feces was evaluated using PCR and qPCR. Information on gender, age, region (province), rural/peri-urban location, and body mass index (BMI) was collected.

### Principal findings

The overall rate of *Blastocystis* carriage was 67.2%. Univariate analysis revealed significant associations between *Blastocystis* carriage and region (p<0.05), location (p<0.001) and age group (p<0.05). Logistic regression analysis revealed that rural/peri-urban location and BMI were significantly associated with *Blastocystis* carriage. Nine subtypes (ST1-ST7, ST10 and ST23) were identified with ST3, ST7 and ST1 as the most abundant ones, in this order. The greatest diversity of subtypes, in terms of numbers, was found in the middle aged group (nine subtypes), while the least diversity was found in the young adult and obese (three subtypes each) groups.

**Data Availability Statement:** Sequence data were submitted to GenBank under accession numbers

PP728964–PP729011, PP729403–PP729456 and PP731515–PP731535.

**Funding:** This work was funded by Thailand Research Fund (RSA6080048) awarded to EG, Thailand Science Research and Innovation (652A01021) award to SP. VJ was funded by a Postdoctoral Fellowship from Mae Fah Luang University (06/2023). The funders had no role in study design, data collection and analysis, decision to publish, or preparation of the manuscript.

**Competing interests:** The authors have declared that no competing interests exist.

## Conclusions

This study increases the understanding of the epidemiology of *Blastocystis* colonization and its association with population parameters and characteristics in middle-income countries.

## Author summary

*Blastocystis* is an enteric microbial eukaryote of ubiquitous, worldwide occurrence in both humans and animals. Based on the small subunit ribosomal RNA gene, *Blastocystis* has been divided into genetically distinct subgroups, called subtypes. It has been hypothesized that some of these subtypes might be pathogenic. Despite a century of research efforts, gaps remain on our knowledge of *Blastocystis* epidemiology. In this study, the authors investigated occurrence of *Blastocystis* among Thai, gut-healthy adults from three provinces in a rural/peri-urban gradient and examined associations with age, area of residence and body mass index (BMI). In this study, 67% of the participants were *Blastocystis* positive. Region, location and age group were individually associated with *Blastocystis* infection. Statistical model testing indicated that location and BMI were also associated with *Blastocystis* carriage. Nine subtypes were identified with the middle aged group having the greatest diversity (all nine subtypes). The obese and young adult groups had the least subtype diversity (three subtypes). This study sheds lights on the molecular epidemiology of *Blastocystis* in middle-income countries.

## Introduction

*Blastocystis* is a common intestinal protist of a wide range of vertebrate and invertebrate animals [1–4]. The role *Blastocystis* plays in human health and disease remains unclear with some studies reporting associations with inflammation and disease, while many others have demonstrated that the organism is associated with overall gut health [5–9]. Comparison of gut-healthy individuals and those exhibiting symptoms showed no link between presence of *Blastocystis* and gastrointestinal disorders, especially irritable bowel syndrome (IBS), though there were some exceptions [10–16]. Moreover, gut-healthy individuals with low-to-normal body mass index (BMI) are reportedly more prone to being colonized by *Blastocystis* [1,17–19].

*Blastocystis* small subunit rDNA (*SSU* rDNA) sequences have been divided into 42 subtypes (STs) in mammalian and avian hosts [20,21,22–29,30]. Sixteen of these have been reported in humans with a strong preponderance, however, of subtypes 1–4 [3,9,29,31–37]. Though it has been hypothesized that the genetic make-up of the various *Blastocystis* subtypes could potentially reflect differences in its public health significance, so far, no clear trends between symptom development and subtype have been identified.

Nonetheless, variation in the prevalence of *Blastocystis* in the human population may mirror demographic, socioeconomic, epidemiological and transmission factors [32,33,38]. Many studies on *Blastocystis* have focused on geographic region, with dissimilarities observed between high-, middle- and low-income countries. Still, the prevalence of *Blastocystis* may vary widely not only between countries, but also between regions within the same country [6,7,39,40]. Human *Blastocystis* colonization in Europe ranges between 25% and 56%, whereas rates of 37%–100% were reported in Asia and Africa [5–7,9,41–45]. Generally, the prevalence and the diversity of *Blastocystis* subtypes in people living in rural areas is higher than those living in urban locations [7,46–52].

Nonetheless, few studies have provided data on the distribution of *Blastocystis* subtypes in humans across both rural and urban regions. Moreover, the prevalence of *Blastocystis* and certain subtypes in relation to BMI is poorly understood, especially in middle-income countries. The aim of the present study was to investigate the positivity rate and distribution of *Blastocystis* in Thai adults from the background population, focusing on identifying an urban-rural gradient and associations between BMI and colonization.

## Methods

### 1. Ethics statement

The Human Ethics Committee of Mae Fah Luang University (Ethics registry number REH60103) and the Human Ethics Committee of Phramongkutklao College of Medicine (License number S053q/58) approved of the collection of human fecal samples. All enrolled participants were informed of the objectives and procedures involved in this study. All participants signed an informed consent form prior to the collection of fecal samples.

### 2. Study sites and sample collection

Two hundred and thirty-eight fecal samples were obtained once from adults with no gastrointestinal symptoms in three provinces: 95, 57, and 86 samples from Chiang Rai (CR), Phayao (PY), and Chachoengsao (CH), respectively (Fig 1). Forty-five samples were from a previous collection (GenBank accession numbers: OL351649–OL351797; 9), while the rest were newly collected. Three locations were sampled in the CR and the CH area, and one village was sampled in the PY area. The inclusion criteria were adult age, while gastrointestinal symptoms at the time of sampling, diagnosed gastrointestinal disease, and antibiotic treatment up to two months prior to sampling were the exclusion criteria.

### 3. Metadata

Metadata available for study included body mass index (BMI), age, gender, location (rural and/or peri-urban) and region (province)(S1 Data). BMI was calculated and assigned categories as follows: 'lean' ($<25.0$ kg/m$^2$), 'overweight' ($25.0–29.9$ kg/m$^2$), and 'obese' ($\geq 30.0$ kg/m$^2$). Participants were also categorized according to the Asia-Pacific guidelines recommendation for Asian populations, which calls for lower cut off points for 'overweight' ($23.0–27.5$ kg/m$^2$), and 'obese' ($> 27.5$ kg/m$^2$) [53,54]. Age was classified as 'young adult', 'adult', 'middle-aged', and 'aged' for the age groups 19–24, 25–44, 45–64 and $\geq$65 years, respectively [55]. The gender categories 'male' and 'female' were used. Finally, a location was considered as rural or peri-urban based on information on population size, type of area, occupation and availability of medical services and basic necessities [51,56], while the regional origin of all volunteers was divided into provinces (Chiang Rai, Phayao and Chachoengsao). All data collected from each subject remained confidential and were fully anonymized through the encryption of the identity of individuals.

### 4. DNA extraction

Total genomic DNA was extracted directly from human feces using QIAamp DNA Stool Mini Kit (Qiagen, Hilden, Germany) according to the manufacturer's recommendations. The DNA was stored at -20˚C until analyzed.

### 5. Molecular detection and differentiation of *Blastocystis*

The samples were not screened for co-infections with other parasites, bacteria or viruses.

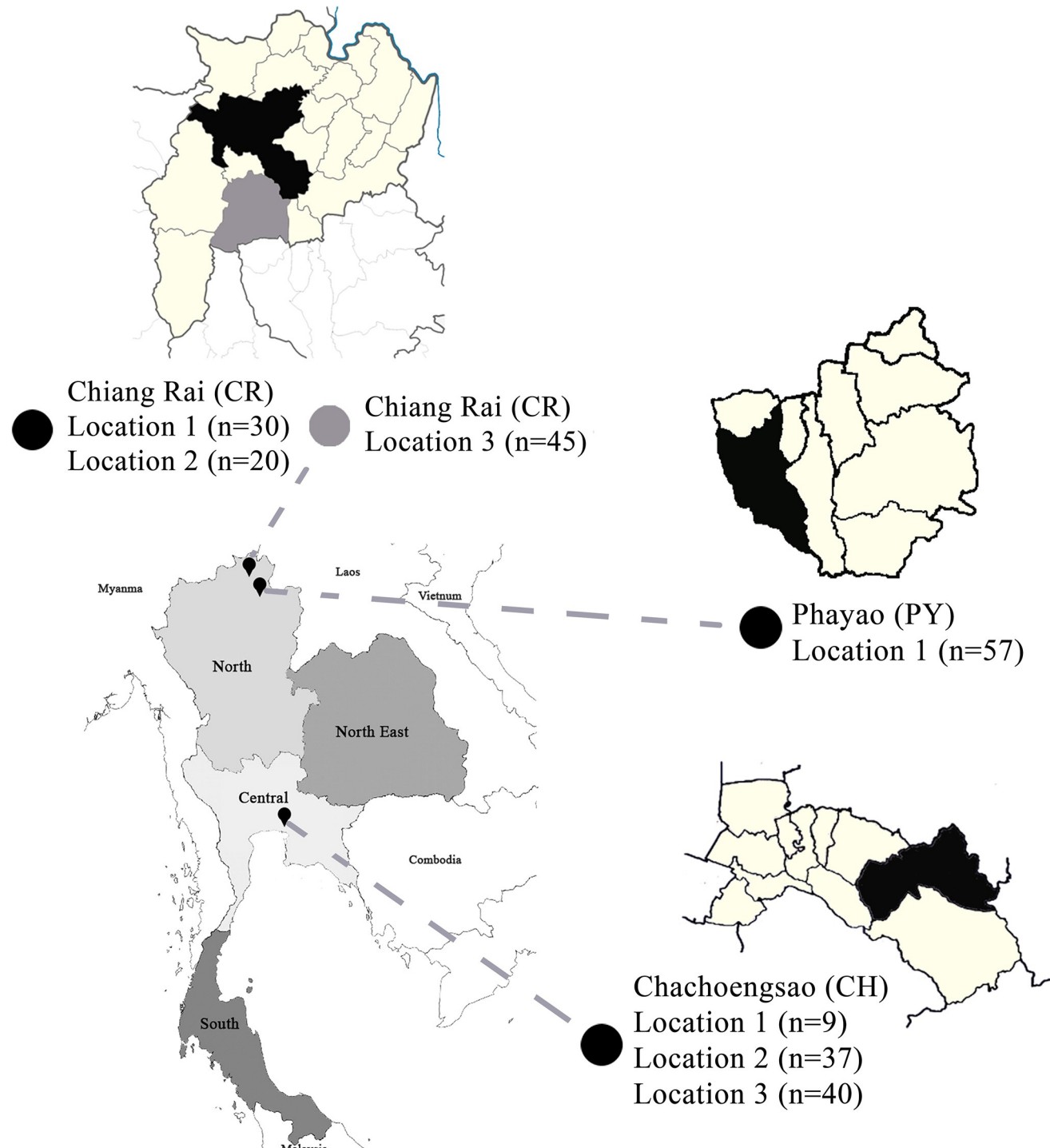

**Fig 1. Study area covered three provinces of Thailand.** CR = Chiang Rai, PY = Phayao, CH = Chachoengsao). Basemap traced from https://earthexplorer.usgs.gov/ courtesy of the U.S. Geological Survey, terms of use https://www.usgs.gov/information-policies-and-instructions/copyrights-and-credits.

The nuclear *SSU* rDNA was used for screening and subtyping purposes. To achieve this, both quantitative polymerase chain reaction (qPCR) and polymerase chain reaction (PCR) were used. The former was employed for *Blastocystis* screening because of its high sensitivity,

however the fragments obtained were of short length (~300 bp) and not always suitable for subtype identification. Hence, conventional PCR was used on all samples to obtain the barcoding region, which comprises a longer fragment and is widely used for subtyping. The approach was also used to compare the results with qPCR.

*qPCR.* qPCR was performed according to standard conditions with primers BL18SPPF1 (5'-AGTAGTCATACGCTCGTCTCAAA-3') and BL18SR2PP (5'-TCTTCGTT ACCCGTTACTGC-3') [57]. The SensiFAST SYBR No-ROX Kit (BIOLINE, UK) was used for all qPCR reactions. qPCR conditions were as previously described [9].

*PCR.* PCR amplification of DNA was performed using a nested PCR reaction. Briefly, the first PCR used the broad-specificity eukaryotic primer RD5 (5'-GGAAGCTTATCTGGTTGA TCCTGCCAGTA-3') and RD3 (5'-GGGATCCTGATCCTTC CGCAGGTTCACCTAC-3'), which amplifies a ~1,800 bp fragment of the *SSU* rDNA [58]. The following reaction was used: initial denaturation for 3 min at 94˚C, 35 cycles with denaturation at 94˚C for 100 s, annealing at 65˚C for 100 s, extension at 72˚C for 100 s, and a final elongation step at 72˚C for 10 min. The first PCR product was used as template in the second PCR reaction (PCR product, ~660 bp) with the *Blastocystis* barcoding primers RD5F (5'-ATCTGGTTGATCCTGCCAGT-3') and BhRDr9R (5'GAGCTTTTTAACTGCAACAACG-3') [59]. The PCR involved initial denaturing at 94˚C for 3 min, followed by 30 cycles including denaturing at 94˚C for 60 s, annealing at 60˚C for 60 s, and extension at 72˚C for 100 s, which was followed by an additional 10-min elongation at 72˚C. The reaction used EmeraldAmp GT PCR Master Mix (TaKaRa Bio USA, Inc.).

## 6. qPCR and PCR product purification and sequencing

All positive PCR and qPCR products were purified using the MEGAquick-spin Plus Total Fragment DNA Purification Kit (iNtRON Biotechnology, Inc., Korea) according to the manufacturer's instructions and sequenced using the reverse (PCR or qPCR) primer (Bionics Company, Korea).

## 7. Cloning

Seven PCR and fifty-five qPCR products for which mixed peak chromatograms were observed were cloned using the pLUG-Prime TA-cloning Vector Kit II (iNtRON, Korea) following the manufacturer's protocol. To maximize the chances of obtaining a clean sequence, two colonies per transformation were screened from each sample and sequenced by Sanger sequencing (Bionics Company, Korea).

## 8. Sequence analysis and subtype identification

The raw reads were edited and ambiguous bases at both ends were removed. The sequences were then used as queries to perform a BLAST search to check whether they were indeed *Blastocystis*. The sequences were then subtyped using the curated PubMLST *Blastocystis* database available online (https://pubmlst.org/organisms/blastocystis-spp). All sequences were submitted to GenBank under accession numbers PP728964–PP729011, PP729403–PP729456 and PP731515–PP731535.

## 9. Statistical analysis

Statistical analyses were carried out using the Stata13 [60](Stata Statistical Software: Release 13. College Station, TX: StataCorp LLC) software program to examine possible association of *Blastocystis* carriage and population parameters. The D'Agostino test was used to determine

the distribution of the continuous parameters measured in this study. Age was not normally distributed; hence, five age groups were developed (see metadata section above). Data were also analyzed by region for regional variations according to gender, age, location, and BMI status using WHO and Asia-Pacific classification criteria ($\chi^2$ test). Raw BMI data were used after logarithmic transformation. Pairwise comparison of means with Bonferroni correction was used to examine associations with the different population parameters. *Blastocystis* carriage and associations with other parameters were examined using univariate and multivariable analysis. For the univariate analysis, data (% proportions) were grouped according to *Blastocystis* carriage and associations with gender, region, location and age were tested. Pairwise comparison of means with Bonferroni correction was used to examine associations with combinations of different population parameters. For the multivariable level analysis, logistic regression models were developed to examine the associations between *Blastocystis* carriage and population parameters. Goodness of fit was established using Akaike information criterion (AIC) and Bayesian information criterion (BIC).

## Results

A total of 238 human fecal samples were collected from 80 (33.6%) and 158 (66.4%) male and female individuals, respectively. The overall age range was 19–82 years. The age distribution was not normal and no transformation was approaching normality. Information on BMI was not available for the 86 participants from the CH province. The BMI range for the remaining study participants was 15.6–37.8 with a geometric mean of 23.87 (95% confidence interval (CI): 23.29–24.48).

Univariate analyses revealed no significant association between sex and regional distribution of the samples ($\chi^2$ = 0.7605, p = 0.684). The distribution of samples by age group was not the same in each region ($\chi^2$ = 19.6746, p<0.001). In CR, the samples from individuals younger than 25 years represented 28.4% of the total in the region. In PY and CH samples from individuals younger than 25 years represented 1.8% and 2.3% of the total in each area, respectively. There was no significant difference in regional BMI, when raw BMI data was used (t = 1.0912, df = 150, p = 0.2769). Using the WHO BMI status, no significant difference was identified between CR and PY ($\chi^2$ = 0.233, p = 0.233). When the Asian BMI status was examined, the difference was significant ($\chi^2$ = 36.0773, p<0.05), with proportionally more lean (LN) individuals identified in the CR province (43.2%) compared to PY (28.1%). The same applied to obese (OB) individuals, 21.1% in CR and 12.3% in CY. Overweight individuals had higher representation in the PY region (59.7%) compared with CR region (35.8%). The majority of the samples originated from rural areas (168, 70.6%). This was consistent across all regions, with PY having no samples from peri-urban locations. The associations of the different population parameters by region are summarized in Table 1.

### *Blastocystis* prevalence and association with variables

Overall, by qPCR, 160 out of 238 samples (67.2%, 95%CI: 60.96–72.93) were positive for *Blastocystis*. When using PCR, the overall detection rate was 47.1% (112/238). Samples positive for PCR were also positive for qPCR. Fifty six PCR products were false positives as evidenced by Sanger sequencing (Fungi, Plantae), while only two qPCR products (Fungi) were false positive. False positives were not included in the calculations determining carriage and subsequent analysis. In terms of the 55 cloned products, 18 were from CR, 27 cases were from PY and 17 cases were from CH locations.

**Univariate analysis.** Males and females had almost equal *Blastocystis* prevalence (68.8% [95%CI: 57.70–78.02] and 66.5% [95%CI: 58.65–73.43], respectively), and hence, no significant

**Table 1. Regional distribution and associations for population parameters, univariate analysis.**

|  |  | CR | PY | CH | Total |
|---|---|---|---|---|---|
|  |  | n (%) | n (%) | n (%) | n (%) |
| Gender | Female | 62 (65.3) | 36 (63.2) | 60 (69.8) | 158 (66.4) |
|  | Male | 33 (34.7) | 21 (36.8) | 26 (30.2) | 80 (33.6) |
|  |  | $\chi^2 = 0.7605$, p = 0.684 | | | |
| Age Group | <24.99 | 27 (28.4) | 1 (1.8) | 2 (2.3) | 30 (12.6) |
|  | 25–44.99 | 17 (17.9) | 4 (7.0) | 21 (24.4) | 42 (17.7) |
|  | 45–64.99 | 36 (37.9) | 42 (73.5) | 40 (46.5) | 118 (49.6) |
|  | >65 | 15 (15.8) | 10 (17.5) | 23 (26.7) | 48 (20.2) |
|  |  | $\chi^2 = 19.6746$, p<0.001 | | | |
| BMI status | LN | 54 (56.8) | 36 (63.2) | ND | 90 (59.2) |
|  | OV | 36 (34.7) | 20 (35.1) | ND | 53 (34.9) |
|  | OB | 8 (8.4) | 1 (1.8) | ND | 9 (5.9) |
|  |  | $\chi^2 = 2.9153$, p = 0.233 | | | |
| Asian BMI Status | LN | 41 (43.2) | 16 (28.1) | ND | 57 (37.5) |
|  | OV | 34 (35.8) | 34 (59.7) | ND | 68 (44.7) |
|  | OB | 20 (21.1) | 7 (12.3) | ND | 27 (17.8) |
|  |  | $\chi^2 = 8.2391$, p<0.05 | | | |
| Residential area | Peri-urban | 30 (31.6) | 0 (0.00) | 40 (46.5) | 70 (29.4) |
|  | Rural | 68 (38.7) | 57 (100.00) | 46 (53.5) | 168 (70.6) |
|  |  | $\chi^2 = 36.0773$, p<0.001 | | | |
|  | Total | 95 (39.92) | 57 (23.95) | 86 (36.13) | 238 |
|  |  | Geometric mean (95% CI) | Geometric mean (95% CI) |  | Geometric mean (95% CI) |
| BMI |  | 23.6 (22.77–24.50) | 24.3 (23.65–24.97) | ND | 23.9 (23.29–24.48) |
|  |  | t = -1.0912, df = 150, p = 0.2769 | | | |

CR = Chiang Rai, PY = Phayao, CH = Chachoengsao; LN = Lean, OV = Overweight, OB = Obese; ND = no data; CI = confidence interval

gender-associated difference in carriage rate was observed ($\chi^2 = 0.1269$, p = 0.722). In terms of region, *Blastocystis* was detected in 61 out of 95 samples in CR (64.2% [95%CI: 54.01–73.27]), 46 out of 57 samples in PY (80.7% [95%CI: 68.21–89.07]), and 53 out of 86 samples in CH (61.6% [95% CI: 50.86–71.37%]), ($\chi^2 = 6.3134$, p<0.05). The median age of participants that were *Blastocystis* positive was 55 years and for the *Blastocystis* negative, 54 years. A significant association was noted between *Blastocystis* colonization and age-related subgroups ($\chi^2 = 9.5192$, p<0.05), with the organism being most common in middle-aged individuals and less common in young adults. Considering location, the prevalence of *Blastocystis* among the people who lived in rural areas was 75.6% (95%CI: 68.48–81.54) in contrast to 47.1% (95%CI: 35.67–68.48) among those who lived in peri-urban areas. People living in rural areas were significantly more likely of carrying *Blastocystis* ($\chi^2 = 18.1554$, p<0.0001). Results are summarized in Table 2.

We tested the association between *Blastocystis* carriage and BMI, where univariate analysis identified a significant association between the two (t = -2.9821, df = 150, p<0.005). The power of the test ($\alpha = 0.05$) was 0.8135 (Table 3). The mean (geometric) BMI for *Blastocystis* positive individuals was 24.5 (95%CI: 23.77–25.15) and for negative 22.56 (95%CI: 21.51–23.67). Following the WHO BMI classification, *Blastocystis* was detected in 58 of 90 (64.4% [95%CI: 53.92–73.73]) of the lean, 41 out of 53 (77.4% [95%CI: 63.97–86.80]) of the overweight and 8 out of 9 (88.9% [95%CI: 46.42–98.66%]) of the obese individuals. However, there was no

**Table 2. *Blastocystis* detection associations with proportions of population parameters, univariate analysis.**

| | | *Blastocystis* negative | | *Blastocystis* positive | | Total | |
|---|---|---|---|---|---|---|---|
| | | n | Proportion % (95%CI) | n | Proportion % (95%CI) | n | Proportion % (95%CI) |
| Sex | Female | 53 | 33.5, (26.57–41.32) | 105 | 66.5, (58.65–73.43) | 158 | 66.4, (60.10–72.14) |
| | Male | 25 | 31.25, (21.98–42.30) | 55 | 68.75, (57.70–78.02) | 80 | 33.6, (29.86–39.90) |
| | | $\chi^2 = 0,1269$, p = 0.722 | | | | | |
| Region | CR | 34 | 35.8, (26.73–45.99) | 61 | 64.2, (54.01–73.27) | 95 | 39.9, (33.84–46.32) |
| | PY | 11 | 19.3, (10.93–31.79) | 46 | 80.7, (68.21–89.07) | 57 | 24.0, (18.92–29.83) |
| | CH | 33 | 38.4, (28.63–49.14) | 53 | 61.6, (50.86–71.37) | 86 | 36.1, (30.24–42.48) |
| | | $\chi^2 = 6.3134$, p<0.05 | | | | | |
| Location | Peri-urban | 37 | 58.9, (41.08–64.33) | 33 | 47.1, (35.67–68.48) | 70 | 29.4, (23.93–35.56) |
| | Rural | 41 | 24.4, (18.46–32.52) | 127 | 75.6, (68.48 81.54) | 168 | 70.6, (64.44–76.07) |
| | | $\chi^2 = 18.1554$, p<0.001 | | | | | |
| Age Group | ≤24.9 | 17 | 51.5, (34.61–68.08) | 16 | 48.5, (31.92–65.39) | 33 | 13.9, (1.00–18.90) |
| | 25–44.9 | 14 | 35.9, (22.34–52.16) | 25 | 64.1, (47.84–66.78) | 39 | 16.4, (12.18–21.69) |
| | 45–64.9 | 29 | 24.6, (17.59–33.22) | 89 | 75.4, (66.78–82.41) | 118 | 49.6, (43.22–55.95) |
| | ≥65 | 18 | 37.50, (24.89–52.07) | 30 | 62.50, (47.93–75.11) | 48 | 20.1, (15.52–25.79) |
| | | $\chi^2 = 9.5192$, p<0.05 | | | | | |
| Total | | 78 | 32.8, (27.07–39.04) | 160 | 67.2, (60.96–72.93) | 238 | 100 |

CR = Chiang Rai, PY = Phayao, CH = Chachoengsao; CI = confidence interval

significant association between *Blastocystis* detection and BMI as classified by the WHO criteria ($\chi^2$ = 4.2393, p = 0.120). Following the Asia-Pacific BMI classification, *Blastocystis* was detected in 33 out of 57 (57.9% [95%CI: 44.61–70.12]) of the lean, 25 out of 33 (75.8% [95%CI: 58.04–87.59]) of the overweight and 49 out of 62 (79% [95%CI: 66.94–87.53]) of the obese individuals. *Blastocystis* carriage was significantly associated with the Asia-Pacific BMI status ($\chi^2$ = 6.9485, p<0.05). Results are summarized in Table 3.

**Multivariable analysis.** Multivariable analysis led to a logistic regression model (n = 152, LR$\chi$2 = 19.41, p<0.001) that identified BMI (OR [odds ratio] for each point:

**Table 3. *Blastocystis* detection associations with BMI and BMI status, univariate analysis.** For raw BMI data the power of the test is also included.

| | *Blastocystis* negative | | *Blastocystis* positive | | Total | |
|---|---|---|---|---|---|---|
| | n | Geometric mean (95%CI) | n | Geometric mean (95%CI) | n | Geometric mean (95%CI) |
| BMI | 45 | 22.6 (21.51–23.67) | 107 | 24.5 (23.77–25.15) | 152 | 23.3 (23.29–24.48) |
| | t = -2.9821, p<0.005, a = 0.05, power = 0.8135 | | | | | |
| | n | % (95%CI) | n | % (95%CI) | n | % (95%CI) |
| BMI LN | 32 | 35.6, (26.27–49.08) | 58 | 64.4, (53.92–73.73) | 90 | 59.2, (51.14–66.82) |
| OV | 12 | 22.6, (13.20–36.03) | 41 | 77.4, (63.97–86.80) | 53 | 34.9, (27.64–42.86) |
| OB | 1 | 11.1, (1.34–53.58) | 8 | 88.9, (46.42–98.66) | 9 | 5.9, (3.09–11.06) |
| | $\chi$2 = 4.2393, p = 0.120 | | | | | |
| ABMI LN | 24 | 42.1, (29.88–55.39) | 33 | 57.9, (44.61–70.12) | 57 | 37.5, (30.09–45.54) |
| OV | 8 | 24.2, (12.41–41.96) | 25 | 75.8, (58.04–87.59) | 33 | 21.7, (15.81–29.06) |
| OB | 13 | 21, (12.47–33.06) | 49 | 79.0, (66.94–87.53) | 62 | 40.8, (33.18–48.86) |
| | $x^2 = 6.9485$, p<0.05 | | | | | |
| Total | 45 | 29.6, (22.82–37.43) | 107 | 70.4, (62.57–77.18) | 152 | 100 |

CR = Chiang Rai, PY = Phayao, CH = Chachoengsao; LN = Lean, OV = Overweight, OB = Obese; ABMI = Asian-Pacific BMI; CI = confidence interval

**Table 4. Logistic regression models of the associations between *Blastocystis* carriage and population parameters.** Goodness of fit was established using AIC and BIC. BMI was examined both as raw data and as BMI status using the WHO and Asian classification.

| z | | OR | SE | z | p | 95%CI |
|---|---|---|---|---|---|---|
| BMI | | 1.142 | 0.0590 | 2.57 | <0.05 | 1.0323–1.264 |
| Rural/Peri-urban | | 4.431 | 1.9985 | 3.30 | <0.005 | 1.831–10.725 |
| | | | | n = 152, LR$\chi^2$ = 19.41, p<0.001 | | |
| WHO BMI | OV | 1.876 | 0.7854 | 1.50 | 0.133 | 0.826–4.262 |
| | OB | 10.7975 | 12.4443 | 2.06 | <0.05 | 1.128–10.359 |
| Rural/Peri-urban | | 5.875 | 2.7623 | 3.77 | <0.001 | 2.338–14.765 |
| | | | | n = 152, LR$\chi^2$ = 19.73, p<0.001 | | |
| ABMI | OV | 1.745 | 0.7275 | 1.34 | 0.182 | 0.771–3.951 |
| | OB | 4.085 | 2.4811 | 2.32 | 0.05 | 1.243–13.433 |
| Rural/Peri-urban | | 4.799 | 2.266 | 3.32 | p.005 | 1.902–12.108 |
| | | | | n = 152, LR$\chi^2$ = 18.62, p<0.001 | | |

OV = Overweight, OB = Obese; ABMI = Asian-Pacific BMI; CI = confidence interval. OR = odds ratio; SE = standard error

1.142; 95%CI: 1.033–1.264) and location type (OR for rural over peri-urban: 4.431 [95%CI: 1.831–10.725]) having significant associations with *Blastocystis* carriage. The Akaike's Information criterion (AIC) and Bayesian information criterion (BIC) were used to estimate goodness of fit. Similar models were fit with the BMI status using the WHO or Asia-Pacific classification. In all models, BMI status and location were significantly associated with *Blastocystis* carriage. In testing the model, age groups were also examined however no interactions were identified and both AIC and BIC favored the model including only the type of location (Table 4, Fig 2).

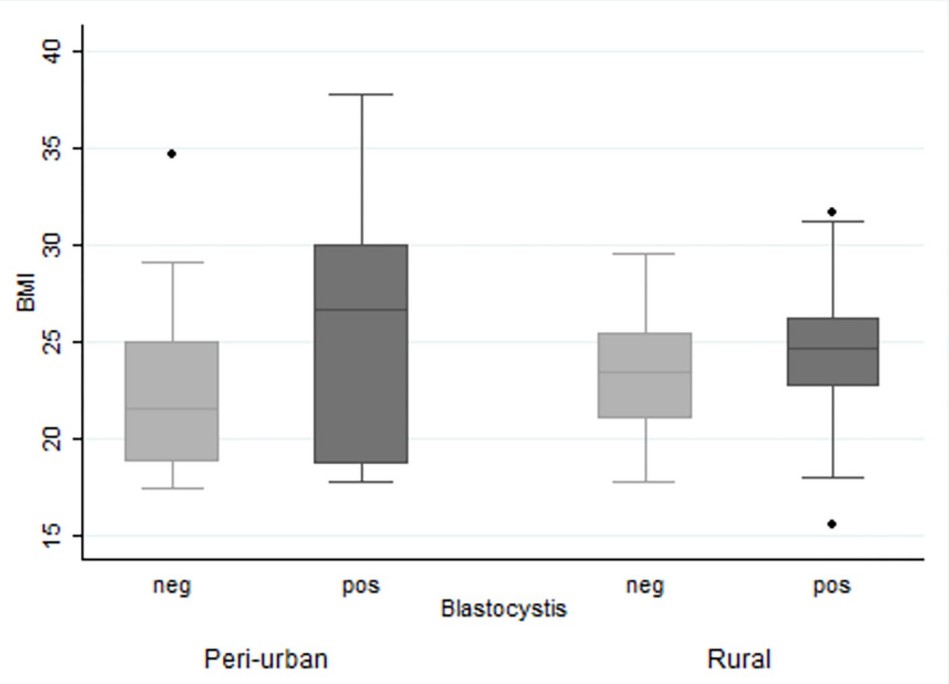

**Fig 2. Boxplot of BMI distribution against *Blastocystis* detection and location type of sample collection.**

**Table 5. Distribution of *Blastocystis* subtypes (ST) according to gender, BMI (WHO classification) and age.**

| Parameters | B+ve | ST1 | ST2 | ST3 | ST4 | ST5 | ST6 | ST7 | ST10 | ST23 | mixST | unk |
|---|---|---|---|---|---|---|---|---|---|---|---|---|
| Male | 55 | 12 | 1 | 26 | 0 | 1 | 1 | 5 | 1 | 2 | 0 | 6 |
| Female | 105 | 11 | 1 | 37 | 1 | 1 | 0 | 32 | 3 | 7 | 6 | 6 |
| Young Adult | 16 | 4 | 0 | 10 | 0 | 0 | 0 | 2 | 0 | 0 | 0 | 0 |
| Adult | 25 | 3 | 1 | 8 | 0 | 0 | 0 | 9 | 0 | 1 | 0 | 3 |
| Middle-aged | 89 | 12 | 1 | 34 | 1 | 1 | 1 | 20 | 2 | 6 | 4 | 7 |
| Aged | 30 | 4 | 0 | 11 | 0 | 1 | 0 | 6 | 2 | 2 | 2 | 2 |
| Lean (LN) | 58 | 8 | 2 | 20 | 0 | 1 | 0 | 10 | 2 | 6 | 0 | 9 |
| Overweight (OV) | 41 | 5 | 0 | 16 | 1 | 0 | 1 | 9 | 1 | 3 | 3 | 2 |
| Obese (OB) | 8 | 3 | 0 | 3 | 0 | 0 | 0 | 1 | 0 | 0 | 0 | 1 |

B+ve = *Blastocystis* positive; unk = unknown

### *Blastocystis* subtype distribution

Altogether, we detected a single subtype in 88.8% (142/160) of *Blastocystis*-positive individuals, whereas a mixed-subtype (mixST) infection was detected in 3.8% (6/160). Nine subtypes were identified. The most abundant subtype was ST3 (n = 63), followed by ST7 (n = 37), ST1 (n = 23), ST23 (n = 9), ST10 (n = 4), ST2 (n = 2), ST5 (n = 2), ST4 (n = 1) and ST6 (n = 1). In 7.5% (12/160) of the cases, the subtype could not be identified precisely either due to short length and/or presence of ambiguous chromatogram peaks (unk).

According to gender, *Blastocystis* ST3, ST1, and ST7 were the most frequently found in males, while ST3, ST7 and ST1 were the most frequently found subtypes in females in that order. *Blastocystis* ST1-ST7, ST10 and ST23 were identified in lean and overweight, whereas in obese participants carriage was limited to ST1, ST3 and ST7 (Table 5). Finally, *Blastocystis* ST1, ST3 and ST7 were the most common subtypes across all age groups. The middle-aged (45–64 years) individuals exhibited the highest diversity of subtypes. Regarding region, *Blastocystis* ST1-ST7, ST10 and ST23 were observed in CR province, ST1-ST3 and ST7 in PY province and ST1, ST3, ST5, ST7 and ST10 in CH province (Fig 3). The top three *Blastocystis* subtypes in CR were ST3, ST1 and ST23, while ST3, ST7 and ST1 were the most abundant in PY and CH province, in this order.

When looking at the distribution of *Blastocystis* subtypes according to type of location, greater diversity was observed in the rural area (ST1-ST7, ST10 and ST23) compared with the peri-urban locations (ST1, ST3, ST5 and ST7) (Fig 4). Moreover, in the CH rural location ST1, ST3, ST7, ST10 were detected, whereas in the CH peri-urban areas ST1, ST3, ST5 and ST7 were found. Lastly, ST1-ST3 and ST7 were characterized in the rural location of PY.

## Discussion

*Blastocystis* is one of the most frequently reported intestinal microeukaryotes in human fecal specimens, but it has also been identified in animals and the environment [2–4,61,62]. Limited data on *Blastocystis* reported from humans indicates a higher prevalence in Africa and Asia [44,45,47,63–65]. Studies in Thailand have reported positivity rates ranging between 14.5% to 73.0% in adults [7,9,66] and 4.8% to 89% in children [63,67–70]. In the present study, the overall *Blastocystis* colonization rate was 67.2% (160/238) using molecular methods in line with our previous work in the area [9]. The reported differences in prevalence across studies are likely due to the variable sensitivity of the methods used. This is also exemplified herein, whereby the detection rate using qPCR was higher than when employing conventional PCR.

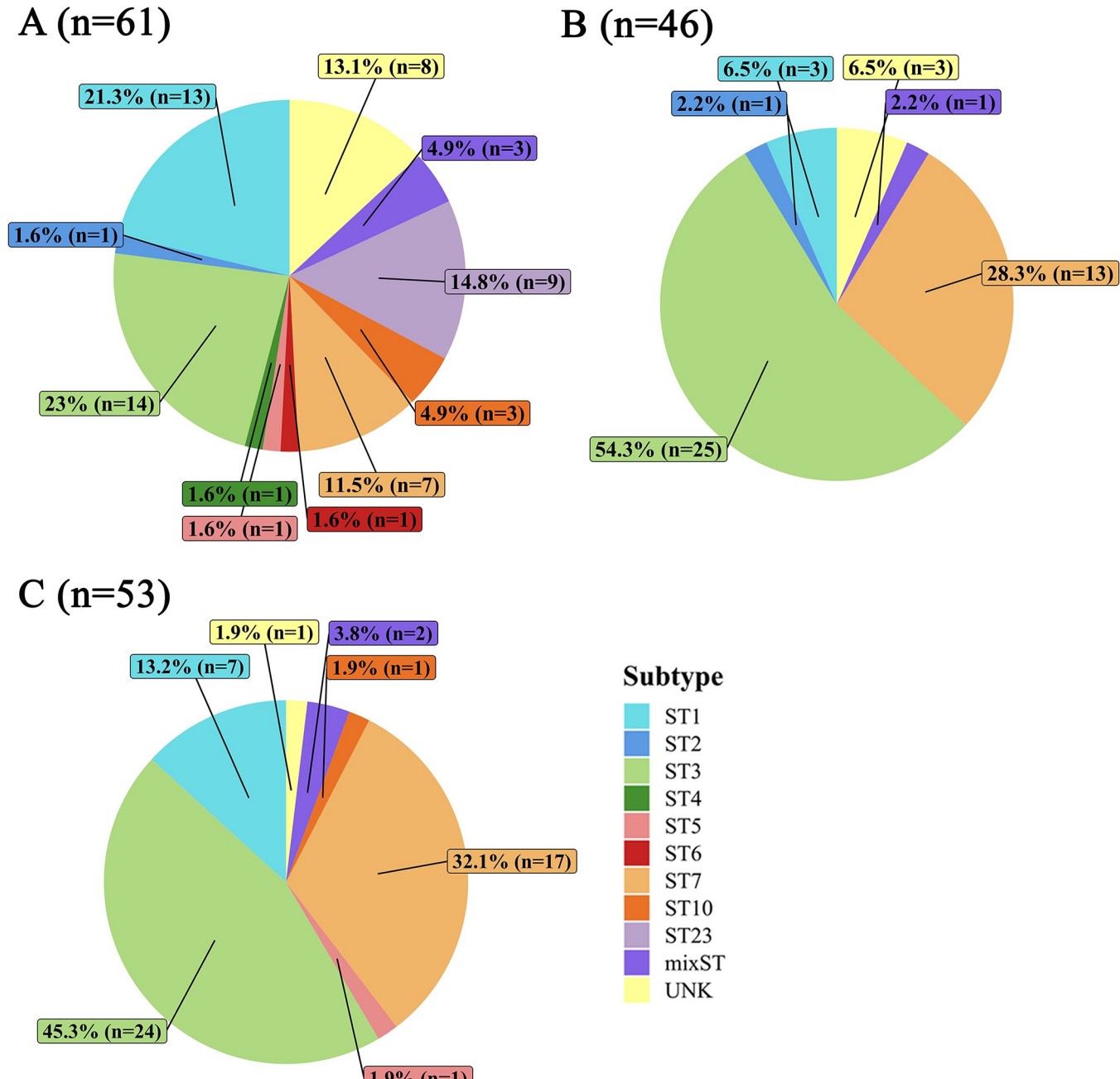

**Fig 3. Distribution of *Blastocystis* subtypes (in percent and raw numbers) among the 160 positive subjects from the three studied provinces.** (A) Chiang Rai (CR). (B) Phayao (PY). (C) Chachoengsao (CH).

Moreover, variability of populations and the demographic factors investigated also likely contributed to the different results.

The prevalence of *Blastocystis* and associations with population parameters was examined. *Blastocystis* carriage was not associated to the gender of the participants in line with earlier observations [6,71–74]. In terms of BMI, there was no association between prevalence and BMI WHO classification in the univariate analysis. However, there was a significant association when considering the Asia-Pacific classification. Applying a logistic regression model

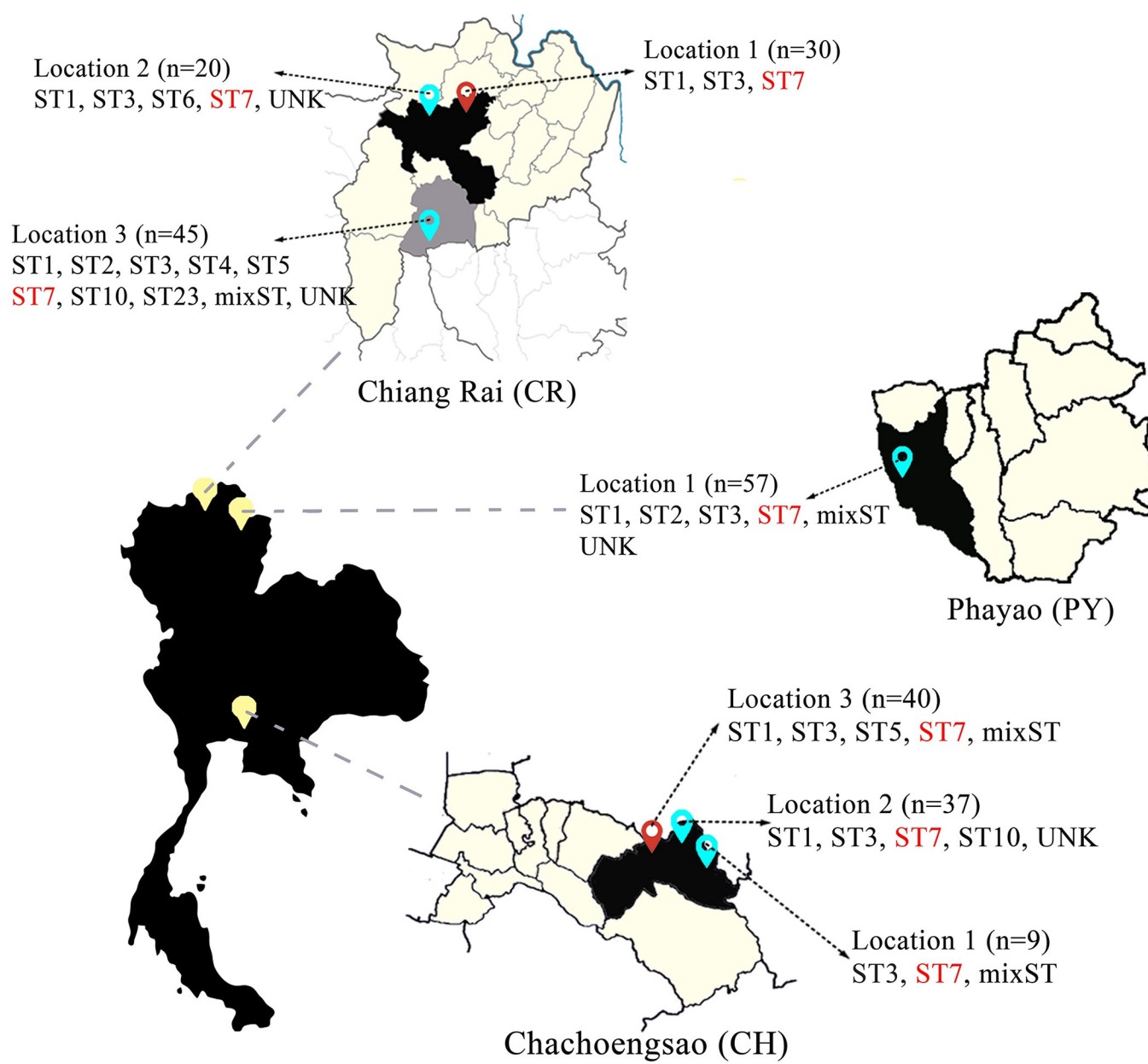

**Fig 4. Distribution of *Blastocystis* subtypes in humans based on sampling locations.** A total of 238 volunteers participated in the study, of which 160 were *Blastocystis*-positive. ST7 was present in every location sampled and is in red font. Basemap traced from https://earthexplorer.usgs.gov/ courtesy of the U.S. Geological Survey, terms of use https://www.usgs.gov/information-policies-and-instructions/copyrights-and-credits.

identified a significant association with BMI (both WHO and Asia-Pacific classification) confirming previous investigations [1,18,19,75]. The higher prevalence of *Blastocystis* in obese rather than lean subjects is contrary to several previous studies. One could speculate that the low number of obese participants may have affected the result though this was not the case when BMI raw data were used in the analysis. Nonetheless, data is emerging that *Blastocystis* carriage rate in obese individuals might be higher than previously thought. Specifically, the organism was found in over 40% of three separate obese populations [76–78].

In the univariate analysis, a significant association between *Blastocystis* carriage and age was detected in line with previous studies [6,8,71,79]. In testing the model, age groups were also examined and, given the variable age distribution in peri-urban and rural areas, age is a possible contributor to the association between location type and *Blastocystis* carriage, yet no interactions were identified and both AIC and BIC favored the model including only the type of location.

*Blastocystis* occurrence in rural areas was significantly higher than in peri-urban locations, aligning with previous findings in Turkey, Czech Republic, Iran and Qatar [8,64,74,80]. Personal and community-level hygiene, geographic area, culture, and lifestyle of a population may play important roles in the distribution and prevalence of the organism.

Nine subtypes were identified including ST1-ST7, ST10 and ST23. The diversity of subtypes was highest in the rural areas; these are locations, where a variety of domesticated animals and wildlife are found in close proximity to humans. The richness of the identified subtypes might be due to the study area, climate, culture, food, and lifestyle. Overall, the top-three dominant subtypes were ST3, ST7 and ST1. While ST1 and ST3 are commonly identified worldwide in humans, ST7 seems to be geographically restricted in Asia [7,9,70]. In vitro and mouse model studies, pointed towards ST7 being pathogenic [81–84]. Nonetheless the pathogenicity of this subtype in humans remains unclear. Hence, the notable finding here is the presence of ST7 in gut-healthy participants, which aligns with results from our previous studies in the area [7,70] and argues against ST7 being pathogenic. On the contrary, it points towards ST7 being a common subtype in this gut-healthy population.

How did so many human hosts in our study acquire ST7? One explanation could be their proximity to avian hosts, well-known carriers of this subtype (85). Raising domestic fowl is routine practice in rural Thailand, while wildlife birds also abound. In turn, fowl could be shedding *Blastocystis* and contaminating soil and water. This would be in line with our previous work in the area, where we identified ST7 in both soil and water (9).

The presence of several ST10 and ST23 sequences is also of interest, as they are not typically found in humans. Although there are some reports of human-origin ST10 and ST23 sequences worldwide, this is uncommon [34,70,86], since these subtypes appear to be found more commonly in ruminants [20]. The high number of ST23 sequences in this study raises some questions. One explanation is that the human specimens were contaminated with ruminant fecal matter or the individuals could have been exposed to a fecal/environmental source containing these subtypes simultaneously. Given that all ST23 sequences were found in the same close-knit community, points towards the latter scenario. Notably, both ST10 and ST23 were also detected in schoolchildren in other communities in northern Thailand [70]. Nonetheless, this is not the first time that unusual subtypes are observed in a non-western country. In a recent study from Vietnam, more than half of the *Blastocystis* subtypes were not typical of humans [86]. Further study of these communities should include temporal sampling to determine, whether these subtypes are specific to the community or transient passengers from the environment.

A single sequence of the geographically restricted ST4 was found. Subtype 4 has been found at very low frequencies (usually one or two samples per study) in Asia [7,66,86–90]. Given that

this is a subtype commonly found in rodents throughout Asia [91–94], it is striking that ST4 is barely found in Asian human hosts. This calls into question the zoonotic potential or lack thereof of certain *Blastocystis* subtypes.

Our study has some limitations. The sample size is relatively modest with some categories being under-represented or absent. For instance, aged individuals from peri-urban locations were absent and BMI data was not available for the CH population. Screening for additional colonies or utilizing high throughput sequencing technologies would have shed light on the extent of mixed infections in this population.

## Conclusions

This is a molecular epidemiological study of *Blastocystis* in Thai adults in rural and peri-urban areas that also examines associations with BMI, age and geographic location (rural/peri-urban). Living in a rural area and BMI are significantly associated with *Blastocysti*s carriage. This study and others bring into question the factors that determine colonization by a specific subtype. For instance, even though birds worldwide carry ST7, we mainly see this subtype in humans living Asia. Likewise with ST4, while rodents worldwide carry it, mainly humans living in Europe seem to carry it.

## Supporting information

**S1 Data. Participant raw data.**
(XLS)

## Acknowledgments

The authors wish to thank all volunteers for their enthusiastic participation. We are also grateful to Abby McCain for her assistance in sample collection and Tanapon Seetarason for his assistance in picture graphics. The authors express their gratitude to Mae Fah Luang University for their kind sponsorship to the Gut Microbiome Research Group. The authors acknowledge the COST Action CA21105-*Blastocystis* under One Health, which is supported by COST (European Cooperation in Science and Technology).

## Author Contributions

**Conceptualization:** Vasana Jinatham.

**Formal analysis:** Eleni Michalopoulou.

**Investigation:** Amara Yowang, Thanakrit Vichasilp, Picha Suwannahitatorn, Anastasios D. Tsaousis.

**Methodology:** Vasana Jinatham, Siam Popluechai, Eleni Gentekaki.

**Resources:** Vasana Jinatham, Siam Popluechai, Eleni Gentekaki.

**Supervision:** Siam Popluechai, Anastasios D. Tsaousis, Eleni Gentekaki.

**Validation:** Vasana Jinatham, Siam Popluechai, Eleni Gentekaki.

**Writing – original draft:** Vasana Jinatham.

**Writing – review & editing:** Christen Rune Stensvold, Siam Popluechai, Anastasios D. Tsaousis, Eleni Gentekaki.

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
