## [Decision Letter · Decision Letter 0]

30 Jan 2024

Dear Eleni Gentetaki,

Thank you very much for submitting your manuscript "Risk factors and colonization of Blastocystis in Thai adults" for consideration at PLOS Neglected Tropical Diseases. As with all papers reviewed by the journal, your manuscript was reviewed by members of the editorial board and by several independent reviewers. In light of the reviews (below this email), we would like to invite the resubmission of a significantly-revised version that takes into account the reviewers' comments.

Please consider the suggestions of the three referees who evaluated your article and provide the necessary corrections and additional information, especially in the material and method sections. However, you are expected to explain your limitations in detail by supporting the discussion section with current literature. You can find detailed information in reviewers' comments.

Kind regards

We cannot make any decision about publication until we have seen the revised manuscript and your response to the reviewers' comments. Your revised manuscript is also likely to be sent to reviewers for further evaluation.

Sincerely,

Funda Dogruman-Al, Ph.D., M.D.

Guest Editor

Ricardo Fujiwara

Section Editor

Dear Authors,

Thank you for your submission of the manuscriot that is title "Risk factors and colonization of Blastocystis in Thai adults" (Manuscript Number PNTD-D-23-01403).

Consider the suggestions of the three referees who evaluated your article and provide the necessary corrections and additional information, especially in the material and method sections. However, you are expected to explain your limitations in detail by supporting the discussion section with current literature. You can find detailed information in reviewers' comment.

Kind regards

Reviewer's Responses to Questions

**Key Review Criteria Required for Acceptance?**

**Methods**

-Are the objectives of the study clearly articulated with a clear testable hypothesis stated?

-Is the study design appropriate to address the stated objectives?

-Is the population clearly described and appropriate for the hypothesis being tested?

-Is the sample size sufficient to ensure adequate power to address the hypothesis being tested?

-Were correct statistical analysis used to support conclusions?

-Are there concerns about ethical or regulatory requirements being met?

Reviewer #1: (No Response)

Reviewer #2: - objectives are clear

- design is adequate

- population is well described

- laboratory methods are adequate. Some questions on the efficacy of mixed infection detection are raised in the detailed comments below.

- for statistical analysis, see comments below

- ethical and regulatory requirements were met.

Reviewer #3: The aims of the study are clearly stated with a clear and testable hypothesis.

The study design was appropriate to meet the stated objectives.

The sample size is sufficient to address the hypothesis being tested.

Proper statistical analysis was used to support the results.

There are no concerns about meeting ethical or regulatory requirements.

**Results**

-Does the analysis presented match the analysis plan?

-Are the results clearly and completely presented?

-Are the figures (Tables, Images) of sufficient quality for clarity?

Reviewer #1: (No Response)

Reviewer #2: - Results are presented adequately, with an exception for one Figure (see below)

Reviewer #3: The analysis presented matches the analysis plan.

The results are presented clearly and completely.

Figures (Tables, Images) are of sufficient quality in terms of clarity.

**Conclusions**

-Are the conclusions supported by the data presented?

-Are the limitations of analysis clearly described?

-Do the authors discuss how these data can be helpful to advance our understanding of the topic under study?

-Is public health relevance addressed?

Reviewer #1: (No Response)

Reviewer #2: Discussion is sober and informative. Some improvements are suggested in the comments below. 

The limitations are apparent, yet not discussed. 

Public health relevance is addressed.

Reviewer #3: Conclusions are supported by the data presented.

The issue of public health is being addressed.

**Editorial and Data Presentation Modifications?**

Reviewer #1: (No Response)

Reviewer #2: Kindly see comments below.

Reviewer #3: Minor Revision

**Summary and General Comments**

Reviewer #1: Comments to Authors

In this manuscript Jinatham et al. describes the occurrence and genetic diversity of the protist of uncertain pathogenicity Blastocystis in an apparently healthy adult population from rural and suburban settings in Thailand. The authors also appraised whether BMI and other sociodemographic variables (age, gender, location) were associated with an increased likelihood of Blastocystis carriage. Detection of the protist was accomplished by molecular (nested-PCR and qPCR) methods, whereas genotyping was conducted by Sanger sequencing of the amplification products obtained by both PCR methods. A thorough statistical analysis involving logistic regression and multiple correspondence analyses were also carried out to assess associations among risk factors and Blastocystis carriage. Overall, the manuscript provides some useful prevalence and molecular data from an endemic area where relatively little information is available on the epidemiology of Blastocystis. However, the study has some design and methodological limitations that must be taken into account by the Authors when interpreting results and raising conclusions. I would also like to bring to the Authors´ attention the convenience of numbering pages and lines of the manuscript to facilitate the review process.

Major issues

1. Sample collection: please indicate the inclusion/exclusion criteria followed when recruiting participants for the study. In my opinion this is a critical step any study design attempting to correlate the presence of a microorganism with the outcome of the infection or any other variable considered.

2. Following this very same line of reasoning, the authors focused on the association Blastocystis vs. BMI. Did they check for the presence of other enteric parasites that might be acting of confounders of the obtained results? If this information is available, I would encourage the Authors to mention it in the manuscript and to reinterpret their results under the light of potential co-infections (expected at high rated in rural areas from endemic countries such as Thailand. If this information is not available, this should be stated as a limitation of the survey. Another important issue that must be acknowledged by the Authors is that BMI data was unavailable for 36% of the participants. This, together with the fact that sample size is not particularly high, might have compromised the quality and robustness of the obtained results.

3. In this regard, it is also unclear to me why the Authors restricted the study to individuals of adult age. If Blastocystis is mainly a commensal (rather than a pathogen) microorganisms in most instances, we should expect increased prevalence rates through lifetime as once the protist colonizes the gut it remains there as part of the microbiome for prolonged period of time. Information from longitudinal studies conducted in human populations in Ireland (see PMID 25077936) and Spain (see PMID 36282323) and in animal populations in USA (see PMID 37264466) seem to support this notion. By focusing only on adult individuals (more likely to carry Blastocystis) I wonder whether the potential effect of primary colonization at younger ages (e.g., paediatric populations) and its effect in BMI is missed. I invite the Authors to explore this possibility more in depth and comment on it. Regarding potential correlation between Blastocystis and obesity, please also see PMIDs 36358463, 35956387, and 34650200. These references can contribute to enrich the quality and interest of the results obtained in this survey.

4. In the Material and Method sections, it is very unclear to me what was the diagnostic algorithm used in the present study. Authors used simultaneously nested-PCR and qPCR. Considering that the latter has more sensitivity and, judging the results obtained here, much better specificity, it seems more reasonable using qPCR as screening method and nested-PCR for genotyping purposes. What was the rationale behind this decision and which benefits were obtained from it?

5. Also, the molecular part of the Material and Methods section is poorly structured and described. I would recommend following this scheme using independent sub-sections:

• DNA extraction and purification

• qPCR protocol

• nested-PCR protocol, including gel electrophoresis and imaging

• Sequencing and sequence analyses

Please note that information provided in these sub-sections should suffice to allow the interested reader to reproduce the experiments without checking primary sources.

6. No GenBank accession numbers are provided, so I was unable to assess the quality of this material. This task must be conducted in a potential second round of review, particularly for less common genetic variants such as ST23. Please provide this information.

7. Discussion section, first paragraph: when comparing results from similar studies conducted in Thailand, I think it is important to provide information regarding the populations surveyed and the detection methods used to enable direct data comparison. Please amend.

8. Discussion section: Authors correctly point out to the animal contamination to explain the origin of ST10 and ST23, but in my opinion they should do the same for ST7 as pigs are recognized hosts for this Blastocystis genetic variant. Please comment on this important matter to highlight the importance of waterborne/foodborne transmission and to give also a One Health perspective to the discussion of this information. Regarding the virulence/pathogenicity of ST7, I think (I am talking from the top of my head and might be wrong) that most the evidence for this statement come form in vivo assays with experimentally infected model animals conducted by Kevin Tan´s group. Please check. If true, this information should be clearly indicated in the manuscript.

9. The Authors must include at the end of the Discussion section a whole paragraph stating the limitations of the study (sample size, potential bias associated to the selected population under study, limited BMI data, absence of co-infection data, etc.). Please be as thorough and precise as possible.

Minor issues

1. Decima figures: Authors use two decimal figures through the text. I believe that a single one is precise enough (e.g., 67.23% should be better expressed as 67.2%. Please modify through text and Tables).

2. Introduction section, second paragraph: SSU (as all gene abbreviations) should be italicised. Amend through the whole manuscript.

3. Introduction section, second paragraph. Please note that at present 42 valid Blastocystis STs are considered valid. ST45 and ST46 have been recently reported in Australian marsupials (see Koehler et al., 2024 https://doi.org/10.1016/j.ijppaw.2023.100902). Also, please note that current references 21-24 do not cover all Blastocystis STs reported. These should also include: 

• ST32: PMID 34504891

• ST35-38: PMID 36677338

• ST39: PMID 36932438

• ST40: PMID 37355198

• ST41: PMID 37195413

• ST42-44: PMID 37658622

• ST45-46: https://doi.org/10.1016/j.ijppaw.2023.100902

Please amend and update this info with the primary references.

4. Introduction section, second paragraph: please note that 16 (not 14) Blastocystis STs have been reported in humans. These include ST1-ST10, ST12, ST14, ST16, ST23, ST35, and ST41. Please see PMIDs 27034056, 32932661, 34956115, 33467077, 34356524, 37195413, and 37658622). Please amend and update this info with the primary references.

5. Cloning subsection: pLUG-Prime® (in superscript). In my opinion, the last three sentences of this section are better suited for the Results section.

Reviewer #2: I read with great interest the manuscript by Jinatham et al who conducted a modestly sized but well-designed epidemiological study on the occurrence, subtypes and predictors of Blastocystis sp. in Thailand, focusing on peri-urban / rural differences, and associations with BMI. 

The design is sound and relevant to the research question. The methods (with one exception) are appropriate and supported by the literature and the experience of the author team. The results are clearly presented, with the exception of one figure. 

The interpretation of the results in Discussion is interesting, mostly reads well and is sober (except for some problems with the interpretation of statistical significance or lack thereof). 

I would like to suggest two points that seem important to me. Addressing them would improve the quality of the manuscript.

(a) Detection methods should be discussed so that the reader understands their purpose. Readers may be confused as to why two PCR methods were used. The rationale behind the combination needs to be briefly explained to readers outside the Blastocystis community. 

(b) I found the method of subcloning the (q)PCR products rather cumbersome and definitely underpowered to detect a mixed infection. While I fully understand that the use of NGS for subtyping a benign parasite in a research setting in a middle-income country may be problematic. I find it difficult to understand why only two colonies were picked after cloning the product. Given that a mixed infection can be detected on Sanger chromatograms from about 20% admixture of the weaker signal, we can simplify to say that the odds of detecting the weaker subtype is 1:4. Two colonies is definitely not enough; even if the signal ratio were 1:1, it would still be too few... The costly thing is the transformation and selection

---

## [Decision Letter · Decision Letter 1]

13 Jun 2024

Dear Eleni Gentekaki

We are pleased to inform you that your manuscript 'Blastocystis colonization and associations with population parameters in Thai adults' has been provisionally accepted for publication in PLOS Neglected Tropical Diseases.

Best regards,

Funda Dogruman-Al, Ph.D., M.D.

Guest Editor

Abhay Satoskar

Section Editor

Dear Authors,

I would like to mention that I have reviewed the revisions made to your manuscript titled "Blastocystis colonization and associations with population parameters in Thai adults, " in response to the suggestions and comments provided by the reviewers.

I am pleased to inform you that the revised manuscript has been thoroughly reviewed and is deemed suitable for publication in our journal.

Thank you for your diligent efforts in addressing the reviewers' feedback. We look forward to seeing your work published.

Kind regards

Reviewer's Responses to Questions

**Key Review Criteria Required for Acceptance?**

**Methods**

-Are the objectives of the study clearly articulated with a clear testable hypothesis stated?

-Is the study design appropriate to address the stated objectives?

-Is the population clearly described and appropriate for the hypothesis being tested?

-Is the sample size sufficient to ensure adequate power to address the hypothesis being tested?

-Were correct statistical analysis used to support conclusions?

-Are there concerns about ethical or regulatory requirements being met?

Reviewer #3: The study design is adequate.

The population is clearly described

Proper statistical analysis was used to support the results.

There are no concerns about meeting ethical or regulatory requirements.

**Results**

-Does the analysis presented match the analysis plan?

-Are the results clearly and completely presented?

-Are the figures (Tables, Images) of sufficient quality for clarity?

Reviewer #3: The analysis presented matches the analysis plan.

Figures (Tables, Images) are of sufficient quality in terms of clarity.

**Conclusions**

-Are the conclusions supported by the data presented?

-Are the limitations of analysis clearly described?

-Do the authors discuss how these data can be helpful to advance our understanding of the topic under study?

-Is public health relevance addressed?

Reviewer #3: Conclusions are supported by the data presented.

The issue of public health is being addressed.

Limitations of the analysis are clearly defined.

**Editorial and Data Presentation Modifications?**

Reviewer #3: (No Response)

**Summary and General Comments**

Reviewer #3: This manuscript has been previously reviewed and the authors have made the changes and corrections requested. It can be published in your journal.

PLOS authors have the option to publish the peer review history of their article (what does this mean?). If published, this will include your full peer review and any attached files.

Reviewer #3: No

---

## [Editor Report · Acceptance letter]

4 Jul 2024

Dear Dr Gentekaki,

We are delighted to inform you that your manuscript, "Blastocystis colonization and associations with population parameters in Thai adults," has been formally accepted for publication in PLOS Neglected Tropical Diseases.

Best regards,

Shaden Kamhawi

co-Editor-in-Chief

Paul Brindley

co-Editor-in-Chief
